# The Effect of Integrative Treatment on Improving Functional Level in Stroke Patients: A Retrospective Chart Review

**DOI:** 10.3390/healthcare13121452

**Published:** 2025-06-17

**Authors:** Daegil Kwon, Sun Hee Ahn, Hyun Jung Jung

**Affiliations:** 1Comprehensive and Integrative Medicine Institute, Daegu 42473, Republic of Korea; cateyesn@naver.com (D.K.); ssuny4363@naver.com (S.H.A.); 2Department of Diagnostics, College of Korean Medicine, Daegu Haany University, Daegu 41072, Republic of Korea

**Keywords:** stroke, functional recovery, integrative treatment

## Abstract

**Objectives:** This study aimed to evaluate the effects of integrative treatment on functional recovery in stroke patients by comparing medical records of those receiving Western rehabilitation treatment only versus integrative treatment in a single hospital. **Methods:** Medical records of 174 stroke patients were analyzed, divided into three groups based on Korean medicinal treatment frequency: Western rehabilitation only (WO), Western rehabilitation with low-frequency Korean medicine (WLK), and Western rehabilitation with high-frequency Korean medicine (WHK). Patients were further categorized into subacute (last 3 months) and chronic (last 6 months) stroke groups. Functional recovery was assessed using tools like the Berg Balance Scale (BBS), Modified Barthel Index (MBI), and others. **Results:** Overall, BBS and MBI scores showed significant improvements in WLK and WHK groups compared to the WO group. In subacute patients at 3 months post-treatment (MPT), BBS scores increased by 1.7 ± 2.0 (WO), 3.3 ± 4.8 (WLK), and 5.6 ± 5.2 (WHK), with significant differences between WO and WHK (*p* < 0.05). In chronic patients at 6 MPT, BBS scores rose by 0.4 ± 1.1 (WO), 1.8 ± 1.7 (WLK), and 5.3 ± 6.4 (WHK), again significant between WO and WHK (*p* < 0.05). MBI scores in subacute patients at 3 MPT increased by 0.7 ± 2.1 (WO), 2.5 ± 2.9 (WLK), and 3.9 ± 5.5 (WHK), with significant differences between WO and WHK (*p* < 0.05). **Conclusions:** Integrative treatment with Korean medicine significantly enhanced balance, daily activity performance, and functional levels in stroke patients compared to Western rehabilitation alone.

## 1. Introduction

Stroke is a major cause of death and a leading cause of disability in adults worldwide [1]. According to a statistical survey in South Korea, the mortality rate due to stroke is decreasing gradually. However, the incidence of stroke has increased with the aging population, leading to increased medical expenses [2]. Approximately 70–75% of stroke patients have serious difficulties in performing essential daily activities such as motor function, gait, and cognitive function, which in turn affect not only the patients themselves but also their family and society [3].

Many stroke patients are often referred to nursing, rehabilitation, and Korean medicine hospitals for treatment to reduce the aftereffects of stroke after acute treatment. One study showed that among stroke patients who undergo Western rehabilitation treatment at rehabilitation hospitals, more than 90% also underwent acupuncture treatment [4]. Acupuncture promotes neurogenesis and cell proliferation, regulates blood flow in ischemic areas, inhibits apoptosis, and modulates neurochemicals involved in the ischemic cascade. Based on these mechanisms, acupuncture is considered effective in enhancing balance and muscle strength and reducing spasticity in stroke patients. It is also thought to aid in the overall improvement of post-stroke aftereffects [5]. In traditional Korean medicine treatment settings, moxibustion therapy, used in combination with acupuncture, has shown significant effects on post-stroke urinary incontinence, dysphagia, and hand edema in patients with hemiplegia [6].

The incidence of chronic diseases has increased in an aging society, leading to an increased demand for East-West integrative medicine treatment, and the East-West Integrative Care System was established in 2010 in South Korea. East-West integrative treatment refers to a treatment method in which clinicians trained in Western or Korean medicinal practices collaborate in the diagnosis and treatment process to complement the weaknesses and strengthen the advantages of both medicinal fields to find a suitable treatment method [7]. It has been recognized as the most realistic and efficient alternative to overcome the limitations of the dual medical system in South Korea [8].

Previous studies have demonstrated that East-West integrative treatment is superior in improving function levels of daily activities and health-related quality of life [9] and is associated with higher patient satisfaction [10] compared to rehabilitation monotherapy in stroke patients. However, these studies were limited in fully understanding the effects of integrative treatment on functional recovery in stroke patients, as they were conducted over a short treatment duration of less than three months or only compared pre- and post-treatment results without considering the frequency or duration of the treatment [9,11]. Therefore, in this study, we analyzed not only the effects of East-West integrative treatment on functional recovery but also the impact of treatment frequency and duration by comparing stroke patients who received East-West integrative treatment for more than three months during inpatient care at a single hospital with those who received only Western rehabilitation treatment.

## 2. Materials and Methods

### 2.1. Study Design

Age, sex, comorbidities, and time elapsed after stroke onset were assessed before the analysis. Depending on the level of Korean medicine treatment received, patients were assigned to the Western rehabilitation treatment only (WO) group, Western rehabilitation treatment plus low-frequency (less than twice a week) Korean medicine (WLK) group, and Western rehabilitation treatment and high-frequency (more than twice a week) Korean medicine (WHK) group. The recovery rate of stroke patients also varied depending on the time elapsed after stroke onset. Therefore, before being assigned to the three treatment groups, patients who had suffered a stroke within the preceding 6 months and more than 6 months ago were classified into subacute and chronic groups, respectively (Figure 1).

### 2.2. Patients

We retrospectively analyzed the medical records of patients who were hospitalized for stroke from December 2015 to July 2019 at a local hospital and received East-West integrated medicine and Western rehabilitation treatment. Patients who were diagnosed with stroke after a computerized tomography scan or magnetic resonance imaging (MRI) and who underwent inpatient treatment for more than 3 months were selected. Patients who were unable to continue treatment due to serious medical conditions, infections during treatment, recurrence, or progression of central nervous system damage were excluded.

### 2.3. Functional Assessment Index

The following indicators were measured in patients at the start of treatment (baseline), 1-month post-treatment (MPT), 3 MPT, and 6 MPT. However, the mini-mental state examination (MMSE) and clinical dementia rating (CDR) scores were measured only at baseline and at 6 MPT.

The Berg Balance Scale (BBS) was used to assess the balance movement ability of patients, and a score of 45 or above is generally considered safe for independent walking [12]. The MBI consists of 10 items necessary for daily life, and each item is divided into five levels. A total score of 0 and 100 indicates inability and complete independence to perform the item, respectively [13]. The muscle strength of the patients was assessed using a manual muscle test (MMT). Six joints on the affected side were measured (shoulder flexion and extension, elbow flexion and extension, wrist flexion and extension, hip flexion and extension, knee flexion and extension, and ankle flexion and extension), and the total MMT score ranged from 0 to 60 points [14]. The upper limb function was evaluated using the manual function test (MFT), and a score of 0 was recorded when no function was observed. The total score ranged from 0 to 32 points [15]. The modified Ashworth scale (MAS) was used to assess joint spasticity and was scored from 0 (no spasticity) to 4 points (severe spasticity with contracture) [16]. In our study, the existing scale of 0, 1, 1+, 2, 3, and 4 points was matched as 0 to 5 points for analysis, and spasticity of the stiffest joint was measured in each patient. Some items of the functional independence measure (FIM) related to movement and gait were modified to measure the function of patients when lying down and while walking. Items related to head control, roll over, sit up, sit to stand, wheelchair to bed, gait, and stair up/down were measured from 0 (complete dependence on others) to 7 points (complete independence) [17]. The modified Brunnstrom classification (MBC) was used for functional evaluation of the fingers, which was evaluated on a 6-point scale from 1 to 6 points based on the movement of the fingers. A score of 6 indicated the ability to throw and receive objects in a nearly normal manner and buttoning and unbuttoning shirts [18]. Monofilament and two-point discrimination tests were performed for sensory evaluation of the patients. Grip strength (in kg) was measured using a dynamometer for the affected hand, and the mean value of the three measurements was used for the analysis. The cognitive function of the patients was evaluated using the Korean MMSE (K-MMSE) and the CDR. A score of 19 or below and 20–23 on the K-MMSE indicated definitive and suspected dementia, respectively [19]. CDR is a tool to assess the severity of dementia and evaluates six domains of function, including memory, orientation, judgment, and problem solving, community affairs, home and hobbies, and personal care [20].

### 2.4. Statistical Analysis

The differences in quantitative and categorical variables between the three groups at baseline were analyzed using one-way analysis of variance (one-way ANOVA) and chi-square tests, respectively. A two-way repeated-measures ANOVA was performed to assess improvements after treatment, and the amount of change between baseline and post-treatment in each variable was compared between the groups using one-way ANOVA. For multiple group comparisons, a post hoc analysis was performed. A paired t-test was performed to analyze the differences in MMSE and CDR scores before and after treatment between the three groups. The SPSS software (IBM Corp. Released 2011. IBM SPSS Statistics for Windows, Version 22.0. Armonk, NY, USA: IBM Corp.) was used for all statistical analyses. All tests were two-sided, and *p*-values < 0.05 were considered statistically significant.

## 3. Results

### 3.1. Patient Characteristics

During the study period, 789 patients were hospitalized for rehabilitation after being diagnosed with a stroke, of which 232, 164, and 219 patients were excluded for less than 90 days of hospitalization, duplication, and re-admission after discharge, respectively. A total of 174 patients, including 83 subacute and 91 chronic patients, were included in this study. Among subacute patients, 84 and 65 patients were followed up to 3 and 6 months, respectively, and among chronic patients, 91 and 59 patients were followed up to 3 and 6 months, respectively. To include a larger number of subjects, patients who had been hospitalized for more than 6 months were included in the analysis at 3 months and 6 months. In subacute patients, there was no difference in general characteristics between the three groups at baseline. However, there was a significant difference in the prevalence of diabetes between the groups. In chronic patients, there was also no difference in general characteristics between the three groups at baseline. However, the size of the major axis of the brain lesion and the urination method were significantly different among the three groups (Table 1, Figure 1).

### 3.2. Effects on Functional Recovery

#### 3.2.1. 3 Months Post-Treatment

Subacute patients: At baseline, there were no significant differences in functional recovery outcomes among the three groups. After 3 months of follow-up, BBS scores increased by 1.7 ± 2.0, 3.3 ± 4.8, and 5.6 ± 5.2 points in the WO, WLK, and WHK groups, respectively. In particular, the WHK group showed a statistically significant increase in the BBS score compared to the WO group. Likewise, MBI scores rose by 0.7 ± 2.1, 2.5 ± 2.9, and 3.9 ± 5.5 points in the WO, WLK, and WHK groups, respectively, with the WHK group showing a significantly greater enhancement than the WO group. Although the WLK group showed numerical gains in BBS and MBI scores, there were no statistically significant differences compared to the WO or WHK groups. MMT, FIM, and MFT scores significantly improved over time in the WHK group; however, there was no significant difference between the groups at 3 MPT (Table 2; Figure 2).

Chronic patients: At baseline, there were no significant differences in most functional recovery outcomes among the three groups. However, several variables showed baseline differences. Specifically, MBI and FIM scores in the WO group were significantly lower than those in the WLK and WHK groups. In addition, baseline MMT and MFT scores in the WLK group were significantly lower than those in the WHK group. After 3 months of follow-up, the BBS score increased by 0.8 ± 1.5, 1.8 ± 2.6, and 2.6 ± 5.6 points in WO, WLK, and WHK groups, respectively. MBI scores increased by 0.5 ± 1.1, 1.3 ± 2.0, and 1.3 ± 2.5 points in the WO, WLK, and WHK groups, respectively, and improved significantly in the WLK and WHK groups. The FIM, MBC, monofilament, two-point discrimination, and MAS scores did not improve significantly in any of the three groups. Moreover, there were no significant differences between the post-treatment groups (Table 3; Figure 2).

#### 3.2.2. 6 Months Post-Treatment

Subacute patients: At baseline, there were no significant differences in functional recovery outcomes among the three groups. After 6 months of follow-up, BBS scores increased by 1.8 ± 2.3, 5.8 ± 8.4, and 8.1 ± 6.1 points in the WO, WLK, and WHK groups, respectively. There was no difference in BBS scores between the three groups at 6 MPT. Both the WLK and WHK groups showed significantly improved BBS scores over time; however, the WO group did not show any such improvement. MBI scores improved significantly across all three groups, with the WHK group showing a 6.1 ± 7.0 point gain, which was significantly greater than the 3.4 ± 3.2 point improvement seen in the WO group. The WLK group showed moderate improvements in both BBS and MBI scores; however, these changes were not statistically significant when compared to the WO or WHK groups. The MMT score improved significantly over time in the WHK group, and the MFT and FIM scores significantly improved over time in the WLK and WHK groups. However, the MMT, MFT, and FIM scores at 6 MPT were not significantly different among the three groups. Moreover, the MBC, monofilament, two-point discrimination, MAS, MMSE, CDR, and dynamometer scores did not change significantly over time in any of the three groups (Table 4; Figure 2).

Chronic patients: At baseline, there were no significant differences in functional recovery outcomes among the three groups. After 6 months of follow-up, BBS scores increased by 0.4 ± 1.1, 1.8 ± 1.7, and 5.3 ± 6.4 points in the WO, WLK, and WHK groups. Only the WHK group showed a significant improvement in BBS scores over time. The increase in BBS scores was significantly greater in the WHK group than in the WO group. MBI scores increased significantly over time in both the WHK and WLK groups, with improvements of 4.7 ± 5.1 and 4.7 ± 7.9 points, respectively, compared to 1.0 ± 2.3 points in the WO group. Although the increase in MBI scores was higher in the WHK group, the difference was not statistically significant. MMT scores increased significantly over time in the WHK and WLK groups; however, the increase in MMT scores at 6 MPT did not differ among the three groups. The MFT score significantly improved over time in all three groups; however, the increase in MFT score at 6 MPT was not significantly different among the three groups. Additionally, FIM and MMSE scores increased significantly in the WHK group; however, the scores at 6 MPT were not significantly different among the three groups. The MBC, monofilament, two-point discrimination, dynamometer, MAS, and CDR scores did not significantly change over time in any of the three groups (Table 5; Figure 2).

## 4. Discussion

Stroke shows different symptoms and recovery rates depending on the area and size of the lesion and the age of the patient. Therefore, habilitation medicine specialists as well as physical therapists, occupational therapists, speech therapists, and other medical professionals participate in the treatment, and Korean medicine treatment can also contribute to the recovery of stroke patients [21].

In this study, the level of improvement of various indicators was assessed and compared in stroke patients who underwent either Western rehabilitation monotherapy or East-West integrative medicine treatment in a single hospital to understand the effects of East-West integrative treatment on functional recovery. At 3 MPT (short-term) and 6 MPT, functional indicators, including BBS and MBI, improved in subacute and chronic stroke patients, and the improvement was greater in the WHK group. Previous studies of subacute stroke patients who had suffered a stroke within the preceding 3 months have also shown that East-West integrative treatment involving acupuncture was effective in improving the MBI score compared to Western rehabilitation treatment alone [9,22].

BBS and MBI scores help evaluate patient balance and ability to perform daily activities, respectively, and were significantly improved in the WHK group in our study. In subacute patients, BBS scores showed a difference of approximately 4 points and 6 points at 3 MPT and 6 MPT, respectively, between the WO and WHK groups. In chronic patients, the difference in BBS scores between the WO and WHK groups was approximately 2 points at 3 MPT and 5 points at 6 MPT. A previous study reported that a change of approximately 4.66 Â points in the BBS score is clinically significant [23]. While statistically significant differences in BBS scores between the WHK and WO groups were observed as early as 3 months post-treatment, the magnitude of change did not exceed the established minimal clinically important difference (MCID) of 4.66 points until the 6-month mark. This suggests that, although early trends may indicate improvement, clinically meaningful functional recovery requires a longer duration of integrated treatment, particularly in patients with chronic stroke. Acupuncture treatment at least twice a week, simultaneous with Western rehabilitation treatment, is thought to improve balance and gait ability in both subacute and chronic stroke patients. Additionally, another study demonstrated that a change of approximately 1.85 points in MBI score is clinically significant [24]. In our study, the MBI score increased by 1.0 ± 2.3 points in the WO group at 6 MPT, which was not statistically significant. However, it increased by 4.7 ± 7.9 and 4.7 ± 5.1 points in the WLK and WHK groups, showing a clinically significant increase.

Like previous studies [25,26,27], we did not observe a significant improvement in the indicators of muscle strength, spasticity, and sensation after treatment. In addition to muscle strength or sensation recovery, various factors may increase the BBS or MBI scores. For example, muscle coordination, visual–spatial perception, cognitive function, balance, proprioception, trunk control, pain (neuropathic or joint pain), and psychological factors are also important in determining the BBS and MBI scores. One hypothesis that supports our findings is that there was no change in the muscle strength and sensation after acupuncture treatment, whereas the overall condition of the patients improved due to the factors listed above. In a 2018 study, a functional brain MRI revealed structural changes in the cerebral cortex after the acupuncture treatment, and the primary areas that showed changes were related to movement, cognition, emotion, and motivation [28]. Other studies also suggest that acupuncture could modulate the functional connectivity in the motor area and affect the disrupted pattern of the whole brain network in stroke patients [29,30]. In addition, animal studies have demonstrated that acupuncture promotes angiogenesis and improves cerebral blood flow, inhibits oxidative stress and inflammation to prevent cell death, and enhances neurogenesis and synaptic plasticity by modulating signaling pathways such as BDNF/VEGF and ERK [31].

This study has several limitations. First, the number of subjects is insufficient because inpatient treatment is limited to cases at a single hospital, leading to a small sample size. Therefore, when patients were classified into 6 groups according to the onset time and treatment frequency, each group ended up with approximately 20 participants, making it difficult to find recovery of the patient’s function have not been identified. Second, this study failed to account for various variables that could influence the functional recovery of patients. Gender has a significant effect on the functional recovery and prognosis of stroke patients. Women are physically weaker than men, so functional recovery is inevitably slow [32]. In addition, the incidence of depression after stroke is more than 78% higher than that of men, which may cause a slow recovery by lowering the quality of life as well as lowering the will to recover [33]. However, in this study, in order to examine the improvement of function recovery, only physical indicators such as balance, daily life index, muscle strength, and rigidity were examined, and psychological variables such as depression and cognitive impairment and emotional support were not examined. Third, this was a retrospective study that analyzed medical records, and as such, potential confounding variables that may affect the functional recovery of stroke patients—such as stroke severity, time since onset, or comorbidities—were not fully controlled. In addition, analysis of the medical records of a single hospital cannot represent the entire patient population, and there may have been a selection bias.

However, this study showed that the group receiving Korean medicine treatments in addition to rehabilitation therapy more than twice a week showed superior effects compared to other groups, particularly demonstrating significant results in the subacute phase within six months rather than in the chronic phase. This can be seen as an opportunity to confirm the synergistic effects of integrative treatment. Further retrospective studies that control for various variables may yield more meaningful results, providing a foundation for establishing a significant treatment model for stroke patients in the future.

## 5. Conclusions

The goal of rehabilitation in stroke patients is to improve their functional levels and restore their daily activity performance. Thus, our findings show that Korean medicine treatment is helpful in the recovery of stroke patients. Previous studies have demonstrated that Korean medicine treatment can significantly increase balance, daily activity performance, and functional level in patients with few side effects compared to other drugs and procedures. Therefore, Korean medicine treatment should be actively considered in the rehabilitation of stroke patients.

## Figures and Tables

**Figure 1 healthcare-13-01452-f001:**
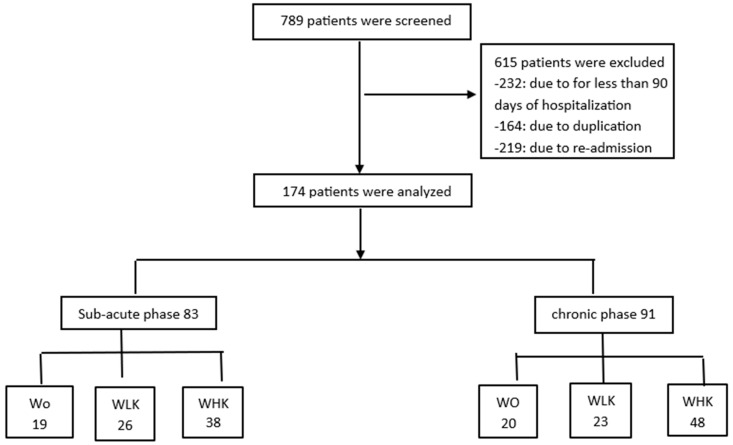
Flow chart for selection. WO, Western rehabilitation treatment only; WLK, Western rehabilitation and low-frequency Korean medicine; WHK, Western rehabilitation and high-frequency Korean medicine.

**Figure 2 healthcare-13-01452-f002:**
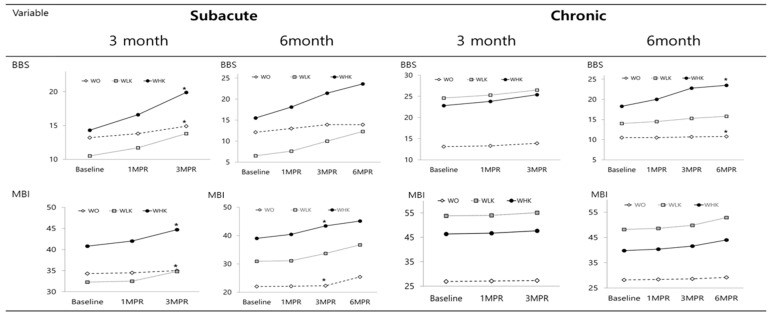
Sequential changes in the Berg balance scale (BBS) and modified Bathel index (MBI) at baseline and months post-treatment (MPT). WO, Western rehabilitation treatment only; WLK, Western rehabilitation and low-frequency Korean medicine; WHK, Western rehabilitation and high-frequency Korean medicine; MPT, months post-treatment. * *p* < 0.05: significant difference in change from baseline to MPT (post hoc Tukey test).

**Table 1 healthcare-13-01452-t001:** General characteristics of patients.

Characteristic	Duration of Onset	Group, *n* or Mean ± SD				*p*-Value
		Total	WO	WLK	WHK	
Number	Subacute	83	19	26	38	
	Chronic	91	20	23	48	
Age (years)	Subacute	71.1 ± 13.5	73.4 ± 11.4	73.6 ± 12.1	68.2 ± 15.0	0.203
	Chronic	62.7 ± 12.9	66.5 ± 13.5	62.7 ± 10.4	61.0 ± 13.7	0.282
Time elapsed after stroke (days)	Subacute	66.2 ± 47.0	68.5 ± 38.7	71.0 ± 45.9	61.8 ± 52.0	0.728
	Chronic	605.7 ± 391.6	619.6 ± 505.6	690.3 ± 373.5	559.3 ± 345.3	0.417
Sex (male–female)	Subacute	30:53	6:13	9:17	15:23	0.827
	Chronic	41:50	10:10	8:15	23:25	0.513
Hypertension (n)	Subacute	37	8	14	15	0.509
	Chronic	36	8	10	18	0.889
Diabetes (n)	Subacute	20	3	11	6	0.032 ^(a)^
	Chronic	22	7	7	8	0.197
Cardiovascular disease (n)	Subacute	16	6	6	4	0.138
	Chronic	8	4		3	0.130
Hyperlipidemia (n)	Subacute	5	1	1	3	0.790
	Chronic	7	1	3	3	0.529
Stroke classification						
Spontaneous intracerebral hemorrhage (n)	Subacute	16	3	3	10	
	Chronic	43	6	12	25	
Traumatic intracerebral hemorrhage (n)	Subacute	9	5	3	1	
	Chronic	6	2	1	3	
Cerebral infarction (n)	Subacute	55	10	19	26	
	Chronic	36	9	10	17	
Stroke site						
ICH (n)	Subacute	16	5	2	9	
	Chronic	37	6	11	20	
SAH (n)	Subacute	6	2	3	1	
	Chronic	7	1	1	5	
SDH (n)	Subacute	3	1	1	1	
	Chronic	3	1	1	1	
ACA (n)	Subacute	2	1	0	1	
	Chronic	0	0	0	0	
MCA (n)	Subacute	40	4	16	20	
	Chronic	28	7	3	13	
PCA (n)	Subacute	0	0	0	0	
	Chronic	1	0	1	0	
Brain stem (n)	Subacute	6	2	2	2	
	Chronic	2	0	1	1	
Cerebellum (n)	Subacute	2	1	1	0	
	Chronic	2	01	0	1	
Small vessel (n)	Subacute	2	2	0	0	
	Chronic	2	1	0	1	
More than 2 vessels (n)	Subacute	2	0	0	2	
	Chronic	1	0	0	1	
Size of brain lesionMajor axis (cm)	Subacute	3.9 ± 2.5	4.5 ± 2.6	4.1 ± 2.9	3.5 ± 2.1	0.486
	Chronic	5.8 ± 3.5	8.2 ± 3.2	3.4 ± 2.0	5.9 ± 3.6	0.002 ^(b)^
Size of brain lesionMinor axis (cm)	Subacute	2.2 ± 13.5	2.4 ± 1.2	2.2 ± 1.8	2.1 ± 1.2	0.789
	Chronic	3.2 ± 1.8	3.7 ± 1.7	2.2 ± 1.1	3.6 ± 2.1	0.071
Urination method						
Self-urination (n)	Subacute	61	14	17	30	
	Chronic	74	11	20	43	
Diaper (n)	Subacute	19	5	8	6	
	Chronic	16	8	3	5	
Catheter (n)	Subacute	3	0	1	2	
	Chronic	1	1	0	0	

^(a)^ *p*-value < 0.05 by chi-square test. ^(b)^ *p*-value < 0.05 by one-way ANOVA. ICH, intra-cerebral hemorrhage; SAH, sub-arachnoid hemorrhage; SDH, sub-dural hemorrhage; ACA, anterior cerebral artery; MCA, middle cerebral artery.

**Table 2 healthcare-13-01452-t002:** Effects of treatment in subacute patients at 3 months post-treatment.

Variable	Group	Time				
		Baseline	1 MPT	Δ1 Month	3 MPT	Δ3 Months
BBS	WO	13.2 ± 16.2 ^(a)^	13.8 ± 16.2 ^(a)^	0.6 ± 1.1	14.9 ± 16.2 ^(a)^	1.7 ± 2.0 ^(c)^
	WLK	10.5 ± 13.4 ^(a)^	11.7 ± 13.5 ^(a)^	1.3 ± 2.0	13.8 ± 13.9 ^(a)^	3.3 ± 4.8
	WHK	14.3 ± 11.8 ^(a)^	16.6 ± 12.6 ^(a)^	2.3 ± 3.7	19.9 ± 12.6 ^(a)^	5.6 ± 5.2 ^(c)^
MBI	WO	34.3 ± 26.3	34.5 ± 26.2	0.2 ± 0.7	35.0 ± 26.1	0.7 ± 2.1 ^(c)^
	WLK	32.3 ± 22.8 ^(a)^	32.5 ± 22.6 ^(a)^	0.2 ± 0.6	34.8 ± 22.9 ^(a)^	2.5 ± 2.9
	WHK	40.8 ± 19.8 ^(a)^	42.0 ± 20.1 ^(a)^	1.2 ± 3.3	44.7 ± 20.2 ^(a)^	3.9 ± 5.5 ^(c)^
MMT	WO	22.9 ± 9.9	23.5 ± 9.5	0.7 ± 1.6	24.0 ± 8.8	1.1 ± 2.3
	WLK	24.4 ± 9.5	25.0 ± 9.4	0.5 ± 2.2	25.3 ± 9.4	0.8 ± 2.8
	WHK	24.9 ± 10.2 ^(a)^	25.1 ± 9.9 ^(a)^	0.2 ± 0.9	25.9 ± 9.6 ^(a)^	0.9 ± 1.6
MFT	WO	15.4 ± 10.6	15.1 ± 10.7	0.1 ± 0.3	15.5 ± 10.4	1.2 ± 2.5
	WLK	12.1 ± 9.7 ^(a)^	12.2 ± 9.8 ^(a)^	0.7 ± 3.9	13.3 ± 9.4 ^(a)^	1.7 ± 4.4
	WHK	11.6 ± 10.6 ^(a)^	12.4 ± 10.3 ^(a)^	0.2 ± 0.7	13.3 ± 10.4 ^(a)^	0.6 ± 1.7
MAS	WO	0.47 ± 0.64	0.47 ± 0.64	0.00 ± 0.00	0.53 ± 0.74	0.07 ± 0.26
	WLK	0.46 ± 0.59	0.50 ± 0.59	0.04 ± 0.20	0.58 ± 0.65	0.13 ± 0.45
	WHK	0.40 ± 0.55	0.40 ± 0.55	0.00 ± 0.00	0.43 ± 0.56	0.03 ± 0.17
FIM	WO	24.2 ± 11.0	24.4 ± 11.1	0.1 ± 0.3	25.4 ± 11.8	1.2 ± 2.5
	WLK	26.0 ± 10.4	26.8 ± 10.5	0.7 ± 3.9	27.7 ± 10.2	1.7 ± 4.4
	WHK	30.6 ± 10.5 ^(a)^	30.8 ± 10.3 ^(a)^	0.2 ± 0.7	31.2 ± 10.5 ^(a)^	0.6 ± 1.7
MBC	WO	5.3 ± 5.3	5.3 ± 1.4	0.0 ± 0.0	5.3 ± 1.4	0.0 ± 0.0
	WLK	5.0 ± 1.8	5.0 ± 1.8	0.0 ± 0.0	5.0 ± 1.8	0.0 ± 0.0
	WHK	3.5 ± 2.2	3.5 ± 2.2	−0.1 ± 0.3	3.5 ± 2.2	−0.1 ± 0.3
MF	WO	3.2 ± 0.4	3.2 ± 0.4	0.0 ± 0.0	3.2 ± 0.4	0.0 ± 0.0
	WLK	3.6 ± 1.0	3.7 ± 1.0	0.1 ± 0.4	3.7 ± 1.0	0.1 ± 0.4
	WHK	3.5 ± 0.9	3.5 ± 0.9	0.0 ± 0.1	3.5 ± 0.9	0.0± 0.1
DNM	WO	7.8 ± 6.0	7.8 ± 6.0	0.0 ± 0.2	7.9 ± 6.0	0.1 ± 0.2
	WLK	7.3 ± 7.2	7.6 ± 7.0	0.3 ± 2.6	8.0 ± 7.4	0.7 ± 3.6
	WHK	7.4 ± 9.4	7.7 ± 9.5	0.3 ± 1.2	7.8 ± 9.5	0.3 ± 1.2

MPT, month post-treatment; WO, Western rehabilitation treatment only; WLK, Western rehabilitation and low-frequency Korean medicine; WHK, Western rehabilitation and high-frequency Korean medicine group; BBS, Berg balance scale; MBI, modified Barthel Index; MMT, manual muscle test; MFT, manual function test; MAS, modified Ashworth scale; FIM, functional independence measure; MBC, modified Brunnstrom classification; MF, monofilament test; DNM, dynamometer. ^(a)^ *p* < 0.05, *p*-value was derived from repeated measures ANOVA for the effect of time. ^(c)^ *p* < 0.05, one-way ANOVA test between the three groups; significant difference between WO and WHK groups by post hoc test.

**Table 3 healthcare-13-01452-t003:** Effects of treatment in chronic patients at 3 months post-treatment.

Variable	Group	Time				
		Baseline	1 MPT	Δ1 Month	3 MPT	Δ3 Months
BBS	WO	13.1 ± 16.6	13.3 ± 16.7	0.2 ± 0.6	13.9 ± 17.0	0.8 ± 1.5
	WLK	24.6 ± 15.1 ^(a)^	25.3 ± 15.2 ^(a)^	0.8 ± 0.8	26.5 ± 15.2 ^(a)^	1.8 ± 2.6
	WHK	22.8 ± 14.4 ^(a)^	23.8 ± 14.0 ^(a)^	1.0 ± 2.4	25.4 ± 14.3 ^(a)^	2.6 ± 5.6
MBI	WO	26.9 ± 21.0 ^(b)(c)^	27.1 ± 21.1	0.2 ± 0.7	27.3 ± 21.2	0.5 ± 1.1
	WLK	53.8 ± 19.3 ^(a)(b)^	54.0 ± 18.9 ^(a)^	0.3 ± 0.8	55.1 ± 18.7 ^(a)^	1.3 ± 2.0
	WHK	46.4 ± 23.6 ^(a)(c)^	46.7 ± 23.3 ^(a)^	0.4 ± 1.8	47.7 ± 23.0 ^(a)^	1.3 ± 2.5
MMT	WO	23.5 ± 11.5	23.7 ± 11.5	0.2 ± 0.6	23.9 ± 11.2	0.4 ± 1.1
	WLK	30.6 ± 5.5 ^(d)^	30.7 ± 5.6	0.2 ± 0.7	30.8 ± 5.6	0.2 ± 0.7
	WHK	25.2 ± 8.3 ^(a)(d)^	25.3 ± 8.2 ^(a)^	0.1 ± 0.4	25.5 ± 8.1 ^(a)^	0.3 ± 0.8
MFT	WO	11.3 ± 10.9 ^(a)^	11.5 ± 10.7 ^(a)^	0.2 ± 0.9	11.9 ± 10.8 ^(a)^	0.6 ± 1.0
	WLK	16.4 ± 8.1 ^(a)(d)^	16.5 ± 8.2 ^(a)^	0.1 ± 0.5	16.9 ± 8.3 ^(a)^	0.5 ± 0.7
	WHK	9.4 ± 8.4 ^(a)(d)^	9.7 ± 8.3 ^(a)^	0.3 ± 1.4	10.0 ± 8.3 ^(a)^	0.6 ± 1.6
MAS	WO	1.21 ± 1.03	1.21 ± 1.03	0.00 ± 0.00	1.16 ± 1.02	−0.05 ± 0.23
	WLK	0.56 ± 0.51	0.56 ± 0.51	0.00 ± 0.00	0.56 ± 0.51	0.00 ± 0.00
	WHK	0.96 ± 0.81	0.93 ± 0.79	−0.02 ± 0.15	0.93 ± 0.79	0.00 ± 0.00
FIM	WO	24.4 ± 13.5 ^(b)(c)^	24.4 ± 13.5	−0.1 ± 0.2	24.6 ± 13.0	0.2 ± 2.8
	WLK	39.4 ± 9.9 ^(b)^	39.4 ± 9.9	0.1 ± 0.2	39.6 ± 9.6	0.3 ± 0.7
	WHK	33.3 ± 11.8 ^(c)^	33.3 ± 11.8	0.1 ± 0.5	33.8 ± 11.8	0.5 ± 1.7
MBC	WO	4.3 ± 2.1	4.3 ± 2.1	0.0 ± 0.0	4.3 ± 2.1	0.0 ± 0.0
	WLK	4.6 ± 1.1	4.6 ± 1.1	0.0 ± 0.0	4.6 ± 1.1	0.0 ± 0.0
	WHK	3.3 ± 2.0	3.3 ± 2.0	0.0 ± 0.0	3.3 ± 2.0	0.0 ± 0.0
MF	WO	3.2 ± 1.5	3.2 ± 1.5	0.0 ± 0.0	3.2 ± 1.5	0.0 ± 0.0
	WLK	3.7 ± 1.0	3.7 ± 1.0	0.0 ± 0.0	3.7 ± 1.0	0.0 ± 0.0
	WHK	3.4 ± 0.8	3.4 ± 0.8	0.0 ± 0.0	3.5 ± 0.8	0.1 ± 0.3
DNM	WO	6.3 ± 10.5	6.2 ± 10.3	−0.1 ± 0.6	6.5 ± 10.7	0.2 ± 0.4
	WLK	8.3 ± 7.3 ^(a)^	8.3 ± 7.3 ^(a)^	0.0 ± 0.2	8.6 ± 7.6 ^(a)^	0.3 ± 0.6
	WHK	8.0 ± 10.7	8.1 ± 10.7	0.0 ± 0.3	8.0 ± 10.7	−0.0 ± 0.5

MPT, month post-treatment; WO, Western rehabilitation treatment only; WLK, Western rehabilitation and low-frequency Korean medicine; WHK, Western rehabilitation and high-frequency Korean medicine group; BBS, Berg balance scale; MBI, modified Barthel Index; MMT, manual muscle test; MFT, manual function test; MAS, modified Ashworth scale; FIM, functional independence measure; MBC, modified Brunnstrom classification; MF, monofilament test; DNM, dynamometer. ^(a)^ *p* < 0.05, *p*-value was derived from repeated measures ANOVA for the effect of time. ^(b)^ *p* < 0.05, one-way ANOVA test between the three groups; significant difference between WO and WLK groups by post hoc test. ^(c)^ *p* < 0.05, one-way ANOVA test between the three groups; significant difference between WO and WHK groups by post hoc test. ^(d)^ *p* < 0.05, one-way ANOVA between the three groups; significant difference between WLK and WHK groups by post hoc test.

**Table 4 healthcare-13-01452-t004:** Effects of treatment in subacute patients at 6 months post-treatment.

Variable	Group				Time			
		Baseline	1 MPT	Δ1 Month	3 MPT	Δ3 Months	6 MPT	Δ6 Months
BBS	WO	12.1 ± 18.2	13.0 ± 18.5	0.9 ± 1.5	13.9 ± 18.3	1.8 ± 2.3	13.9 ± 18.3	1.8 ± 2.3
	WLK	6.5 ± 11.0 ^(a)^	7.6 ± 11.3 ^(a)^	1.2 ± 1.9	10.0 ± 12.5 ^(a)^	3.5 ± 5.5	12.3 ± 13.9 ^(a)^	5.8 ± 8.4
	WHK	15.5 ± 11.7 ^(a)^	18.1 ± 12.8 ^(a)^	2.6 ± 4.3	21.4 ± 12.1 ^(a)^	5.9 ± 5.3	23.6 ± 12.1 ^(a)^	8.1 ± 6.1
MBI	WO	22.0 ± 23.6 ^(a)^	22.1 ± 23.5 ^(a)^	0.1 ± 0.3	22.3 ± 23.3 ^(a)^	0.3 ± 0.6 ^(c)^	25.4 ± 23.7 ^(a)^	3.4 ± 3.2
	WLK	30.9 ± 21.2 ^(a)^	31.1 ± 21.0 ^(a)^	0.2 ± 0.6	33.7 ± 21.5 ^(a)^	2.8 ± 2.9	36.7 ± 21.8 ^(a)^	5.8 ± 5.4
	WHK	39.0 ± 20.2 ^(a)^	40.4 ± 20.7 ^(a)^	1.4 ± 3.6	43.4 ± 21.0 ^(a)^	4.4 ± 5.9 ^(c)^	45.1 ± 21.3 ^(a)^	6.1 ± 7.0
MMT	WO	22.6 ± 10.3	23.4 ± 9.8	0.8 ± 1.7	24.0 ± 9.0	1.4 ± 2.5	24.6 ± 8.7	24.6 ± 8.7
	WLK	24.3 ± 9.3	24.9 ± 9.2	0.6 ± 2.5	25.2 ± 9.4	0.9 ± 3.2	25.6 ± 9.2	25.6 ± 9.2
	WHK	23.3 ± 10.5 ^(a)^	23.6 ± 10.1 ^(a)^	0.4 ±1.0	24.5 ± 10.0 ^(a)^	1.2 ± 1.9	24.6 ± 9.3 ^(a)^	24.6 ± 9.3
MFT	WO	13.6 ± 9.1	13.1 ± 9.2	−0.5 ± 1.7	13.8 ± 8.6	0.2 ± 2.4	14.3 ± 9.2	0.8 ± 2.8
	WLK	10.5 ± 9.7 ^(a)^	10.6 ± 9.8 ^(a)^	0.1 ± 0.3	11.7 ± 9.4 ^(a)^	1.1 ± 2.1	12.9 ± 9.6 ^(a)^	2.3 ± 4.7
	WHK	9.8 ± 10.1 ^(a)^	10.6 ± 9.8 ^(a)^	0.8 ± 2.5	11.6 ± 10.1 ^(a)^	1.8 ± 3.2	12.1 ± 10.1 ^(a)^	2.3 ± 3.7
MAS	WO	0.55 ± 0.69	0.55 ± 0.69	0.00 ± 0.00	0.64 ± 0.81	0.09 ± 0.30	0.64 ± 0.81	0.09 ± 0.30
	WLK	0.48 ± 0.60	0.52 ± 0.60	0.05 ± 0.22	0.62 ± 0.67	0.14 ± 0.48	0.62 ± 0.67	0.14 ± 0.48
	WHK	0.41 ± 0.59	0.41 ± 0.59	0.00 ± 0.00	0.46 ± 0.60	0.05 ± 0.21	0.46 ± 0.60	0.05 ± 0.21
FIM	WO	25.2 ± 11.2	25.3 ± 11.4	0.2 ± 0.4	26.2 ± 11.7	1.0 ± 2.6	26.5 ± 11.8	1.3 ± 3.5
	WLK	25.3 ± 10.3 ^(a)^	25.3 ± 10.7 ^(a)^	0.0 ± 2.1	26.4 ± 10.5 ^(a)^	1.1 ± 2.6	27.1 ± 10.9 ^(a)^	1.9 ± 3.5
	WHK	28.8 ± 11.1 ^(a)^	29.0 ± 10.9 ^(a)^	0.3 ± 0.8	29.5 ± 11.1 ^(a)^	0.8 ± 2.0	30.8 ± 11.2 ^(a)^	2.0 ± 3.3
MBC	WO	4.6 ± 1.7	4.6 ± 1.7	0.0 ± 0.0	4.6 ± 1.7	0.0 ± 0.0	4.6 ± 1.7	0.0 ± 0.0
	WLK	4.7 ± 2.0	4.7 ± 2.0	0.0 ± 0.0	4.7 ± 2.0	0.0 ± 0.0	4.7 ± 2.0	0.0 ± 0.0
	WHK	3.1 ± 2.2	3.0 ± 2.1	−0.1 ± 0.3	3.0 ± 2.1	−0.1 ± 0.3	3.1 ± 2.0	0.0 ± 0.5
MF	WO	3.1 ± 0.5	3.1 ± 0.5	0.0 ± 0.0	3.1 ± 0.5	0.0 ± 0.0	3.4 ± 0.7	0.3 ± 0.6
	WLK	3.9 ± 1.0	3.9 ± 1.0	0.0 ± 0.0	3.9 ± 1.0	0.0 ± 0.0	3.9 ± 1.0	0.0 ± 0.1
	WHK	3.4 ± 0.6	3.4 ± 0.6	0.0 ± 0.0	3.4 ± 0.6	0.0 ± 0.0	3.4 ± 0.6	0.0 ± 0.0
DNM	WO	7.9 ± 6.7	8.0 ± 6.7	0.1 ± 0.2	8.1 ± 6.7	0.1 ± 0.3	9.0 ± 6.6	1.1 ± 1.3
	WLK	7.4 ± 7.4	7.8 ± 7.1	0.3 ± 3.0	8.3 ± 7.6	0.8 ± 4.1	9.0 ± 7.4	1.5 ± 3.9
	WHK	8.6 ± 10.7	8.6 ± 10.7	0.0 ± 0.0	8.7 ± 10.7	0.0 ± 0.2	8.9 ± 10.7	0.3 ± 1.0
K-MMSE	WO	15.6 ± 9.7					13.5 ± 9.7	
	WLK	20.4 ± 7.3					20.8 ± 6.9	
	WHK	20.1 ± 9.3					20.5 ± 9.8	
CDR	WO	1.2 ± 1.1					1.6 ± 1.4	
	WLK	0.9 ± 0.8					0.9 ± 0.8	
	WHK	1.2 ± 1.3					1.1 ± 1.3	

MPT, month post-treatment; WO, Western rehabilitation treatment only; WLK, Western rehabilitation and low-frequency Korean medicine; WHK, Western rehabilitation and high-frequency Korean medicine group; BBS, Berg balance scale; MBI, modified Barthel Index; MMT, manual muscle test; MFT, manual function test; MAS, modified Ashworth scale; FIM, functional independence measure; MBC, modified Brunnstrom classification; MF, monofilament test; DNM, dynamometer; K-MMSE, Korean mini-mental state examination; CDR, clinical dementia rating. ^(a)^ *p* < 0.05, *p*-value was derived from repeated measures ANOVA for the effect of time. ^(c)^ *p* < 0.05, one-way ANOVA between the three groups; significant difference between WO and WHK groups by post hoc test.

**Table 5 healthcare-13-01452-t005:** Effects of treatment in chronic patients at 6 months post-treatment.

Variable	Group				Time			
		Baseline	1 MPT	Δ1 Month	3 MPT	Δ3 Months	6 MPT	Δ6 Months
BBS	WO	10.5 ± 17.2	10.5 ± 17.2	0.0 ± 0.0	10.7 ± 17.2	0.3 ± 1.0	10.8 ± 17.2	0.4 ± 1.1 ^(c)^
	WLK	14.0 ± 13.3	14.5 ± 13.9	0.5 ± 0.6	15.3 ± 13.9	1.3 ± 1.0	15.8 ± 15.0	1.8 ± 1.7
	WHK	18.3 ± 14.0 ^(a)^	20.0 ± 13.6 ^(a)^	1.8 ± 3.3	22.8 ± 14.7 ^(a)^	4.5 ± 7.6	23.5 ± 13.3 ^(a)^	5.3 ± 6.4 ^(c)^
MBI	WO	28.2 ± 22.7	28.4 ± 23.0	0.2 ± 0.8	28.6 ± 23.4	0.5 ± 1.1	29.2 ± 24.2	1.0 ± 2.3
	WLK	48.2 ± 18.8 ^(a)^	48.6 ± 18.3 ^(a)^	0.4 ± 0.9	49.8 ± 18.4 ^(a)^	1.6 ± 2.2	52.9 ± 18.5 ^(a)^	4.7 ± 7.9
	WHK	39.8 ± 25.4 ^(a)^	40.4 ± 25.2 ^(a)^	0.5 ± 2.2	41.6 ± 24.8 ^(a)^	1.8 ± 3.0	44.0 ± 24.6 ^(a)^	4.1 ± 5.1
MMT	WO	24.8 ± 10.7	25.0 ± 10.6	0.2 ± 0.6	25.0 ± 10.6	0.1 ± 0.6	24.1 ± 10.1	−0.8 ± 3.4
	WLK	29.0 ± 5.3 ^(a)^	29.3 ± 5.4 ^(a)^	0.3 ± 0.9	29.4 ± 5.6 ^(a)^	0.4 ± 1.0	30.2 ± 5.5 ^(a)^	1.2 ± 1.5
	WHK	24.5 ± 7.9 ^(a)^	24.6 ± 7.8 ^(a)^	0.1 ± 0.4	24.9 ± 7.8 ^(a)^	0.4 ± 0.7	25.2 ± 7.7 ^(a)^	0.7 ± 1.2
MFT	WO	12.1 ± 12.0 ^(a)^	12.4 ± 11.7 ^(a)^	0.3 ± 1.1	12.9 ± 11.8 ^(a)^	0.8 ± 1.1	12.7 ± 11.8 ^(a)^	0.6 ± 1.3
	WLK	14.2 ± 8.6 ^(a)^	14.4 ± 8.7 ^(a)^	0.1 ± 0.5	14.7 ± 8.8 ^(a)^	0.5 ± 0.9	15.2 ± 9.0 ^(a)^	1.0 ± 1.4
	WHK	8.1 ± 7.3 ^(a)^	8.5 ± 7.1 ^(a)^	0.4 ± 1.8	8.9 ± 7.2 ^(a)^	0.8 ± 1.8	9.4 ± 7.4 ^(a)^	1.3 ± 2.0
MAS	WO	1.2 ± 1.0	1.2 ± 1.0	0.0 ± 0.0	1.1 ± 0.9	−0.1 ± 0.3	1.1 ± 0.9	−0.1 ± 0.3
	WLK	0.7 ± 0.5	0.7 ± 0.5	0.0 ± 0.0	0.7 ± 0.5	0.0 ± 0.0	0.5 ± 0.5	−0.2 ± 0.4
	WHK	1.1 ± 0.9	1.0 ± 0.9	0.0 ± 0.2	1.0 ± 0.9	0.0 ± 0.2	1.0 ± 0.9	0.0 ± 0.2
FIM	WO	27.4 ± 13.6	27.4 ± 13.6	−0.1 ± 0.3	27.3 ± 13.2	−0.1 ± 3.1	27.0 ± 13.6	−0.4 ± 3.3
	WLK	35.2 ± 11.1	35.3 ± 11.2	0.1 ± 0.3	35.6 ± 10.8	0.5 ± 0.9	36.4 ± 10.6	1.2 ± 2.4
	WHK	30.7 ± 11.1 ^(a)^	30.9 ± 11.2 ^(a)^	0.2 ± 0.7	31.7 ± 11.2 ^(a)^	1.0 ± 2.2	32.0 ± 10.7 ^(a)^	1.2 ± 1.7
MBC	WO	4.2 ± 2.5	4.2 ± 2.5	0.0 ± 0.0	4.2 ± 2.5	0.0 ± 0.0	4.2 ± 2.5	0.0 ± 0.0
	WLK	4.4 ± 1.5	4.4 ± 1.5	0.0 ± 0.0	4.4 ± 1.5	0.0 ± 0.0	4.4 ± 1.5	0.0 ± 0.0
	WHK	2.8 ± 2.0	2.8 ± 2.0	0.0 ± 0.0	2.8 ± 2.0	0.0 ± 0.0	2.8 ± 2.0	0.0 ± 0.0
MF	WO	3.6 ± 1.5	3.6 ± 1.5	0.0 ± 0.0	3.6 ± 1.5	0.0 ± 0.0	3.6 ± 1.5	0.0 ± 0.0
	WLK	3.8 ± 0.8	3.8 ± 0.8	0.0 ± 0.0	3.8 ± 0.8	0.0 ± 0.0	3.9 ± 0.7	0.1 ± 0.3
	WHK	3.5 ± 1.0	3.5 ± 1.0	0.0 ± 0.0	3.5 ± 1.0	0.0 ± 0.0	3.4 ± 1.0	0.0 ± 0.1
DNM	WO	8.6 ± 12.3	8.4 ± 12.1	−0.2 ± 0.7	8.7 ± 12.5	0.1 ± 0.3	8.7 ± 12.3	0.1 ± 0.2
	WLK	6.0 ± 7.7	6.1 ± 7.7	0.1 ± 0.3	6.2 ± 8.0	0.2 ± 0.5	7.3 ± 8.1	1.4 ± 2.5
	WHK	7.7 ± 10.3	7.7 ± 10.3	0.0 ± 0.0	7.5 ± 10.2	−0.2 ± 0.5	7.8 ± 10.2	0.1 ± 0.6
MMSE	WO	16.6 ± 11.6					16.4 ± 11.9	
	WLK	15.1 ± 11.8					16.1 ± 10.2	
	WHK	19.9 ± 7.8 ^(e)^					20.8 ± 7.8 ^(e)^	
CDR	WO	1.5 ± 1.5					1.5 ±1.5	
	WLK	0.8 ± 1.1					1.0 ±1.0	
	WHK	1.1 ± 0.8					1.0 ±0.9	

MPT, month post-treatment; WO, Western rehabilitation treatment only; WLK, Western rehabilitation and low-frequency Korean medicine; WHK, Western rehabilitation and high-frequency Korean medicine group; BBS, Berg balance scale; MBI, modified Barthel Index; MMT, manual muscle test; MFT, manual function test; MAS, modified Ashworth scale; FIM, functional independence measure; MBC, modified Brunnstrom classification; MF, monofilament test; DNM, dynamometer; K-MMSE, Korean mini-mental state examination; CDR, clinical dementia rating. ^(a)^ *p* < 0.05, *p*-value was derived from repeated measures ANOVA for the effect of time. ^(c)^ *p* < 0.05, one-way ANOVA between the three groups; significant difference between WO and WHK groups by post hoc test. ^(e)^ *p* < 0.05, *p*-value was derived from a paired *t*-test.

## Data Availability

The data presented in this study are available from the corresponding author upon request. The data are not publicly available due to privacy issues.

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
