# Peer review of "The Effect of Integrative Treatment on Improving Functional Level in Stroke Patients: A Retrospective Chart Review"

_healthcare, 2025, doi:10.3390/healthcare13121452_

Round 1

Reviewer 1 Report

Comments and Suggestions for Authors

The manuscript addresses an important clinical question regarding the effects of integrative treatment on functional recovery in stroke patients. While the study design and findings are presented well, some improvements and queries need to be addressed while revising the manuscript.

  • In the introduction, include some background about Korean medicine treatments, which come under 'integrative treatment' in this study. Are these treatments also beneficial for any other neurological disease?
  • In the Methods section (Study Design), add a flow chart to illustrate the patient classification and group assignment process based on stroke phase and Korean medicine frequency.
  • In Results 3.1, the authors describe patient selection and exclusion in the text, but including a flowchart of the patient selection process would be helpful for easier visualization.
  • The sample size is relatively small when comparing the groups. How does this limitation affect the ability to draw conclusive outcomes?
  • Stroke shows sex-specific differences, with males generally at higher risk than females. This factor could influence the present results and should be considered in the interpretation of the findings.
  • In Figure 1, include the parameter on the Y-axis in the graphs (e.g., BBS or MBI score) for clarity. Additionally, include the full form of MPR in the figure legend.
  • Could the improvement in balance and daily activities be primarily attributed to neurological recovery (i.e., cortical reorganization), or could other factors like pain management or psychological effects have played a role?
  • The study suggests that the data may be useful for future treatment, but there is no clear direction for future research.

Reviewer 2 Report

Comments and Suggestions for Authors

The article addresses a relevant topic—integrative treatment combining Western and Korean medicine for stroke rehabilitation. It highlights the dual medical system in South Korea and the potential of integrative approaches to overcome its limitations, which provides a useful context for readers unfamiliar with the healthcare system. However, the introduction lacks a clear hypothesis or specific research question. The rationale for combining Western and Korean medicine (e.g., acupuncture) is briefly mentioned but not sufficiently supported with prior evidence or a theoretical framework. The literature review appears limited, with few citations provided in the visible excerpts to justify the study’s focus.

The study is a retrospective chart review, which is appropriate for exploring real-world clinical outcomes. The use of standardized outcome measures such as the Berg Balance Scale (BBS), Modified Barthel Index (MBI), Functional Independence Measure (FIM), and Modified Ashworth Scale (MAS) is a strength, as these are validated tools for assessing functional recovery in stroke patients. Moreover, the article does not clearly describe the inclusion/exclusion criteria, sample size, or how patients were assigned to the WO (Western-only) and WHK (Western plus Korean medicine) groups. Without this information, it is difficult to assess the risk of selection bias or confounding factors.

The document mentions spasticity measurement and FIM items but lacks detail on how data were extracted from charts, who performed the assessments, and whether there was blinding to group allocation during data collection.  Additionally, the statistical methods are not described in the provided excerpts. For example, it is unclear whether the study accounted for baseline differences between groups or used appropriate statistical tests to compare outcomes. The repeated text in the document (e.g., "Similarly, MBI scores increased significantly in the WO group") suggests potential errors in editing or data reporting.

The retrospective nature of the study makes it prone to confounding variables (e.g., stroke severity, time since stroke, or comorbidities), but there is no mention of how these were addressed.

The article provides quantitative data on functional outcomes (e.g., BBS and MBI scores) for the WO and WHK groups, with some statistically significant improvements reported in the WHK group. The table on page 9 is detailed, showing mean scores and changes over time for multiple outcome measures.

The table on page 9 includes multiple groups (WO, WHK, WLK), but the WLK group is not explained in the text. This creates confusion about the study design and group definitions

While statistical significance is noted, the clinical meaningfulness of the reported changes (e.g., BBS score differences of 2–6 points) is only briefly discussed in reference to a prior study’s threshold (4.66 points). More discussion is needed to interpret these findings for clinical practice.

The discussion is superficial and does not adequately address the study’s limitations, such as the retrospective design, potential biases, or lack of randomization. It also fails to compare the findings with other integrative treatment studies beyond acupuncture.

Round 2

Reviewer 1 Report

Comments and Suggestions for Authors

The author has addressed all the queries and updated the manuscript accordingly. I recommend it for publication.

Reviewer 2 Report

Comments and Suggestions for Authors

The authors have provided satisfactory explanations to the questions or comments raised